# Pharmacophore modeling and QSAR analysis of anti-HBV flavonols

**Basireh Baei[1], Parnia Askari[2], Fatemeh Sana Askari[3], Seyed Jalal Kiani[4], Alireza Mohebbi** [ID][3,4] *

**1** Infectious Disease Research Center, Golestan University of Medical Sciences, Gorgan, Iran, **2** Department of Life and Science, York University, Toronto, Ontario, Canada, **3** Vista Aria Rena Gene Inc., Gorgan, Golestan, Iran, **4** Department of Virology, School of Medicine, Iran University of Medical Sciences, Tehran, Iran

* alirezaa2s@gmail.com

**Data Availability Statement:** All data underlying our study's findings are freely available. The data are included in the manuscript and Supporting information files.

## Abstract

Due to its global burden, Targeting Hepatitis B virus (HBV) infection in humans is crucial. Herbal medicine has long been significant, with flavonoids demonstrating promising results. Hence, the present study aimed to establish a way of identifying flavonoids with anti-HBV activities. Flavonoid structures with anti-HBV activities were retrieved. A flavonol-based pharmacophore model was established using LigandScout v4.4. Screening was performed using the PharmIt server. A QSAR equation was developed and validated with independent sets of compounds. The applicability domain (AD) was defined using Euclidean distance calculations for model validation. The best model, consisting of 57 features, was generated. High-throughput screening (HTS) using the flavonol-based model resulted in 509 unique hits. The model's accuracy was further validated using a set of FDA-approved chemicals, demonstrating a sensitivity of 71% and a specificity of 100%. Additionally, the QSAR model with two predictors, *x4a* and *qed*, exhibited predictive solid performance with an adjusted-$R^2$ value of 0.85 and 0.90 of $Q^2$. PCA showed essential patterns and relationships within the dataset, with the first two components explaining nearly 98% of the total variance. Current HBV therapies tend to fail to provide a complete cure, emphasizing the need for new therapies. This study's importance was to highlight flavonols as potential anti-HBV medicines, presenting a supplementary option for existing therapy. The QSAR model has been validated with two separate chemical sets, guaranteeing its reproducibility and usefulness for other flavonols by utilizing the predictive characteristics of *X4A* and *qed*. These results provide new possibilities for discovering future anti-HBV drugs by integrating modeling and experimental research.

## Introduction

Hepatitis B virus (HBV) is a significant worldwide health problem because it can lead to acute and chronic liver infections. [1, 2]. HBV is a DNA virus with partial double-stranded characteristics and a member of the *Hepadnaviridae* family. The global health impact is significant, affecting more than 300 million individuals with chronic HBV (CHB) [3] infections and causing approximately 820,000 deaths each year from complications such as cirrhosis and

**Funding:** The author(s) received no specific funding for this work.

**Competing interests:** The authors have declared that no competing interests exist.

hepatocellular carcinoma [4] One major obstacle HBV presents is its ability to persist in the host's hepatocytes, which can result in chronic infection after an initial acute infection. [5–9]. It is also associated with miscarriage and premature birth in pregnant women who are infected. [10, 11]. In the long term, CHB condition raises the chance of severe liver diseases, such as liver cirrhosis and liver failure, and a higher probability of developing hepatocellular carcinoma (HCC). HBV mainly spreads through contact with infected blood or bodily fluids, posing a risk for healthcare workers, drug users, and those having unprotected sex [12]. More-over, HBV can be passed from mother to child during childbirth. [8, 9] HBV can deceptively stay symptomless for long periods, causing gradual immune-mediated damage to the liver. Its detection is limited to the sensitivity of the currently available techniques. This feature, com-bined with the potential for transmission by asymptomatic carriers, highlights the crucial need for effective prevention, early detection, and control of HBV infections worldwide.

Available HBV treatment strategies currently focus on inhibiting viral replication, decreas-ing liver inflammation, and avoiding future complications [3, 13, 14]. Nucleoside analogs (such as lamivudine and tenofovir) and nucleotide analogs (such as entecavir) are essential components of HBV treatment—these antiviral drugs function by blocking the reverse tran-scriptase enzyme, which is necessary for HBV DNA replication. Pegylated interferon-alpha and other interferon-based treatments provide both antiviral and immunomodulatory bene-fits. Recently, there have been promising developments with direct-acting antivirals (DAAs) like Bulevirtide (Myrcludex-B) [15–18] and capsid assembly modulators (CAMs) [19–22] that are progressing in different clinical trial phases, targeting specific stages in the HBV life cycle. The development of innovative therapeutic advancements in treating CHB infection is primar-ily driven by the need to overcome several challenges that hinder the achievement of successful and persistent virological responses with current FDA-approved medications. Accordingly, the emergence of drug resistance is a primary expected concern in viral infection, mainly when nucleos(t)ide analogs are used for an extended period [23]. Patients may need continu-ous treatment throughout their lives to ensure that the virus remains suppressed, which may result in challenges with adherence and possible side effects [24]. Interferon-based treatments frequently lead to flu-like symptoms, tiredness, and depression [25], reducing tolerability. Therefore, there is still a crucial demand for practical treatment approaches that are strong, safe, and able to achieve long-lasting virological responses without the emergence of resistance. In this regard, nature always offers potential sources of antiviral chemicals that have yet to be discovered.

Flavonoids are a variety of natural substances abundant in fruits, vegetables, and medicinal plants. [26]. These compounds are being noticed for their possible antiviral effects, displaying potential in blocking different phases of the HBV life cycle, as we have reviewed before [27]. Research has shown that flavonoids can disrupt viral entry, replication, and assembly, making them promising options for antiviral treatment. Various flavonoid subclasses are introduced with varying mechanisms of action against HBV. Accordingly, Quercetin is distinguished among flavonoids for its capacity to block the production of HBsAg and HBeAg, essential indi-cators of HBV replication. [28–30]. Moreover, Baccharis species like *B. spicata* contain 5-caf-feoylquinic acid (5-CQA) and 3,5-dicaffeoylquinic acid (3,5-DCQA), which have shown antiviral effects against HBV [31]. These compounds exhibit the potential to decrease viral DNA synthesis and reproduction, making them possible contenders for additional investiga-tion in creating new antiviral treatments for HBV. Ongoing research in this field reveals that the wide variety of flavonoids discovered in the natural world could offer a valuable resource of possible antiviral treatments for fighting against HBV and other viral infections.

Due to the diversification of flavonoids from different herbal species, they possess various pharmacological activities that might be preserved among flavonoids from the same subclass

of other plant species. In this regard, a pharmacophore-based model was generated with 100% specificity to discover novel flavonoids with potent anti-HBV activities. Furthermore, flavonoids with antiviral activities against HBV are used to establish a robust and reliable QSAR model for predicting the probable activities of novel compounds. The present study's findings can also be used to create stand-alone applications and highlight plants with potent flavonoids from diverse herbariums.

## Materials and methods

### Retrieving flavonoids' structures

The flavonoid compounds experimentally approved for their anti-HBV activities were reviewed before [27], and the 3D structures of them were obtained from chemical databases, PubChem (https://pubchem.ncbi.nlm.nih.gov/) [32] and ChEMBL (https://www.ebi.ac.uk/chembl) [33].

### Establishing pharmacophore model

The flavonol-based pharmacophore model was created with LigandScout v4.4 [34] as described before [35]. The compounds were grouped into the flavonoid subclasses as reviewed before [27]. Accordingly, a set of nine flavonols with anti-HBV activities, including Kaempferol [29, 36], Isorhamnetin [36], Icaritin [37], Hexamethoxyflavone (Hex) [38], Chrysoeriol-6-C-b-D-boivinopyranosyl-41-O-b-D-glucopyranoside [39], Hyperoside (quercetin-3-O-galactoside) [40, 41], Quercetin-3-O-glucuronide (Q3G), [28, 29, 36], and Myricetin 3-rhamnoside [30] It was used to train the model.

Furthermore, eight flavones (Luteolin 7-O-glucuronide [42], Isovitexin [43], Isoorientin [44–47], Swertisin [48], 4K [49], and Robustaflavone [50]), three flavanones ((-)-Epigallocatechin-3-gallate [51–56], oolonghomobisflavan C (OHBF-C) [57], and sikokianin A [58]), one anthocyanin (proanthocyanidin [57]), one chalcones (rosmarinic acid [59]), one biflavonoid ([60]), and one isoflavone (compound 8f [61]) were used to test and validate the model. Further polyphenols and triterpenes with anti-HBV activities, including PHAP [62], Nobiletin, Linalool [63], Betulinic acid [64], Ursolic acid [65], Taraxasterol [66], and Solamargine [67] were also included as decoys.

The compounds were clustered according to the pharmacophore RDF-code similarity measure with maximum cluster distance calculation methods. The conformers were generated using the software iCon best setting to develop the flavon-based pharmacophore model. Briefly, the maximum number of conformers was set to 200, with an energy window of 20.0 and a max pool size of 4000. The model was created based on pharmacophore fit and atom overlap scoring function. Also, the pharmacophore type was based on Merged Feature Pharmacophore to ensure each feature is scored and those that do not match all input molecules are removed. Further settings were kept as its default. The best-developed model based on the higher score was adopted for virtual screening.

### Screening large chemical database

According to our previous study on screening chemicals in an extensive database of natural and herbal products for the discovery of novel antiviral compounds [68], the PharmIt server (https://pharmit.csb.pitt.edu/search.html) was used for high-throughput screening (HTS) of the flavonol-derived pharmacophore model. In this regard, eleven built-in libraries comprised 1,652,702,330 conformations of 347,839,756 compounds, including CHEMBL32, ChemDiv, ChemSpace, Enamin, MCULE, MCULE_ULTIMATE, MolPort, NCI_Open, PubChem, WuXi

LabNetwork, and ZINC were screened. No filter was applied to the screening process. Moreover, the hit that matched all the pharmacophore features was preserved to provide a library for assessing the precision of the model.

## Creating chemical libraries and assessing model precision

Two datasets of chemical libraries were made to evaluate the accuracy of the established flavonol-based pharmacophore model. The relatively fitted hits (with diverse RMSD values) from the PharmIt screening results were employed to establish the first library set of compounds relatively matched to the model. As reported before, the hits from the built-in datasets might be duplicated. The duplications were removed with the Open Babel command line as described before [69]. The second decoy set was derived from the previous study consisting of 1.7K Lipinski's rule of five-filtered FDA-approved drugs to evaluate the specificity of the model [70]. The accuracy is reported in a receiver operating characteristic (ROC) curve.

## Establishing a predictive QSAR equation

The same combinations were used to develop the pharmacophore model to establish a reliable predictive 2D quantitative structure-activity relationship (QSAR). Accordingly, RDKit (http://www.rdkit.org/), Mordred [71], and Dragon [72] tools were used to calculate 202, 1348, and 2489 descriptors for each molecule, respectively, in the setting of the UseGalaxy server (https://usegalaxy.eu/) [73, 74]. The calculated descriptors were checked to identify and remove duplications and constant or near-constant variables. The rest of the descriptors were retained to predict flavonol compounds' biological activity ($IC_{50}$ μM) and the response variable. The response biological activity variable was extracted from the literature. The activity of compounds that were reported as $g.L^{-1}$ was converted to μM according to the following formula:

$$\mu M = \left( \frac{\mu g}{ML} / MW \right) \times 1000$$

Where μM is the concentration in micromolar, μg/mL is the concentration in micrograms per milliliter, and MW is the predicted molecular weight.

Microsoft Excel (MS-Excel) v2019 was used to perform the QSAR modeling with multiple linear regression (MLR) as we have reported before [75, 76]. Accordingly, the model was built using a 95% confidence interval (CI) and a tolerance level of 0.0001. The model was validated using two sets of test observations: random samples of non-flavonol flavonoids and non-flavonoid compounds. This method assessed the model's replicability by conducting two separate runs of the multiple linear regression (MLR) test with the same explanatory variables.

Selecting the model involved finding the best model based on the adjusted $R^2$. The number of predictor variables in the models was considered within a range of 2 to 4. HC0 was utilized to apply covariance adjustments through the Newey West (adjusted) method of heteroscedasticity. Cook's distance threshold $> 1$ was used to identify outliers or leverage observations in the statistical model. The established model was also tested on anti-HBV flavonols rather than those used for model training. The Goodness of Fit statistics, including coefficient of determination ($R^2$), adjusted $R^2$, mean squared error (MSE), root mean squared error (RMSE), and mean absolute percentage error (MAPE). Furthermore, the squared cross-validated coefficient determined the consistency between the predicted and actual observations in the external flavonol set ($Q^2$).

## Multicollinearity measurement

To evaluate multicollinearity in the QSAR model, the Variance Inflation Factor (VIF) was calculated among the independent variables. The following formula was used to calculate the VIF

values for each included predictor.

$$VIF_i = \frac{1}{1 - R_i^2}$$

Where $R_i^2$ represents the coefficient of determination achieved by performing a regression of the $i^{th}$ independent variable on all other independent variables. If VIF was greater than 10, it was considered as a multicollinearity. This was carried out to ensure the stability and credibility of the model's predictions.

## Principal component analysis

The training flavonols set and selected predictors were used for PCA analysis. This was carried out to reduce the dimensionality of data, visualize connections between variables, and identify trends in compounds through their molecular properties. The exact molecular weight (Exact-MolWt) was also incorporated as a dummy variable [76]. This indicated the weight of each molecule in the data set. A PCA was carried out using Pearson correlation to investigate the relationship between variables. Additionally, the variables were not standardized before conducting PCA. This means that the original scale of the variables was maintained during the PCA analysis. The principal components (PCs) were subjected to Varimax rotation with Kaiser normalization. Five PCs were identified within the dataset to offer an overview of the underlying framework. The findings consisted of a symmetrical biplot that displayed the connections between compounds and features in one graph. The coefficients of the biplot were automatically calculated for visualization. Furthermore, a Bootstrap (on 50 samples) observation chart was developed to assess the reliability of the PCA results.

## Applicability domain (AD) assessment

The Euclidean distance ($d$) was used to measure the straight-line distance between two points, which was used to quantify the similarity between the compounds not seen by the model by considering their properties as coordinates in a mathematical space. A smaller Euclidean distance indicates higher similarity between compounds, which is essential for assessing the AD of the pharmacophore model. $d$ was determined for each compound in both the training and external prediction sets using MS-Excel v2019 using the formula described below. A threshold for determining the validity range was calculated by summing the mean $d$ in the training set and the standard deviation multiplied by 2. This limit represents the maximum distance a compound is judged to be within the model's AD. The $d$ value of each chemical in the external prediction set was also calculated. If a compound's distance surpassed the threshold, it was categorized as *an 'Outlier';* otherwise, it was labeled as an *'Inlier'*.

$$d = \sqrt{\left(Predicted\ IC_{50} - Observed\ IC_{50}\right)^2}$$

*Threshould = Mean of d + (2 × Std.dev of d)*

## Results

### Flavonol-based pharmacophore models

Among the generated pharmacophore models, the best one with a score of 130.7671 was selected (Fig 1). The model comprised 57 features, including one hydrophobic center, three

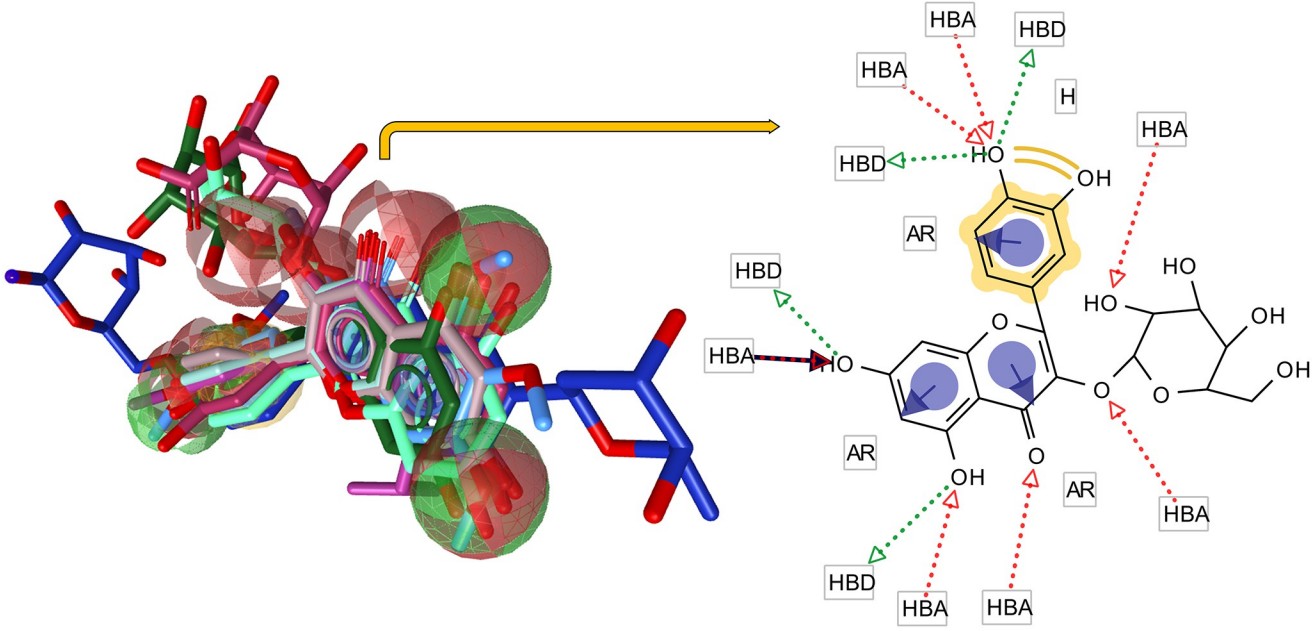

**Fig 1. The flavonol-derived pharmacophore model.** On the left, the compounds are shown in alignment with the pharmacophore. A schematic representation of the alignment is illustrated in the right panel. HBA, Hydrogen-bond donor; HBA, HB acceptor; AR, aromatic ring; H, Hydrophobic center. The exclusive volumes are not shown for clarity of the figure.

aromatic rings, 7 H bond acceptor regions (HBA), 4 H bond donors, and 42 exclusive volumes. A different range of conformers (1 to 200) was generated for each ligand to find the best-matched conformation. The mean pharmacophore score for the training set was 121.01 ± 11.96 (minimum and maximum of 103.4 and 132.54, respectively). Also, the score for the validation set was 49.35 ± 31.36 (minimum of 30.23 and maximum of 106.24). The established model was used for HT screening.

## Virtual screening

The flavonol-based model was deployed for high-throughput screening (HTS) in the PharmIt database, resulting in 1,032 hits. Fig 2 shows that more hits were found in the PubChem dataset. After removing repeated entries, 509 distinct hits were discovered and used to establish a library of active compounds to evaluate the model's precision. Another set of FDA-approved chemicals was used to confirm the model's accuracy. The validation process assumed that the library from HTS primarily had active compounds, which were then matched with the pharmacophore model using an algorithm that only chose those with a strong pharmacophore feature correspondence. On the other hand, even though the FDA database may not have enough flavonoids for specific model comparison, it was still used to test the model's overall performance as a decoy set (see S2 File).

According to Fig 3, the model's validity was assessed against a set of decoys comprised of a library of Drugbank compounds (inactive) using ROC curve analysis to see if the model can distinguish between active flavonol, by which the model is built and inactive compounds or not. Accordingly, the findings showed a sensitivity of 71% and a specificity of 100%, indicating a good discriminatory power. The Area Under the Curve (AUC) values highlighted the model's discriminative ability, 1.00, 1.00, 1.00, and 0.85 for cutoffs of 1%, 5%, 10%, and 100%,

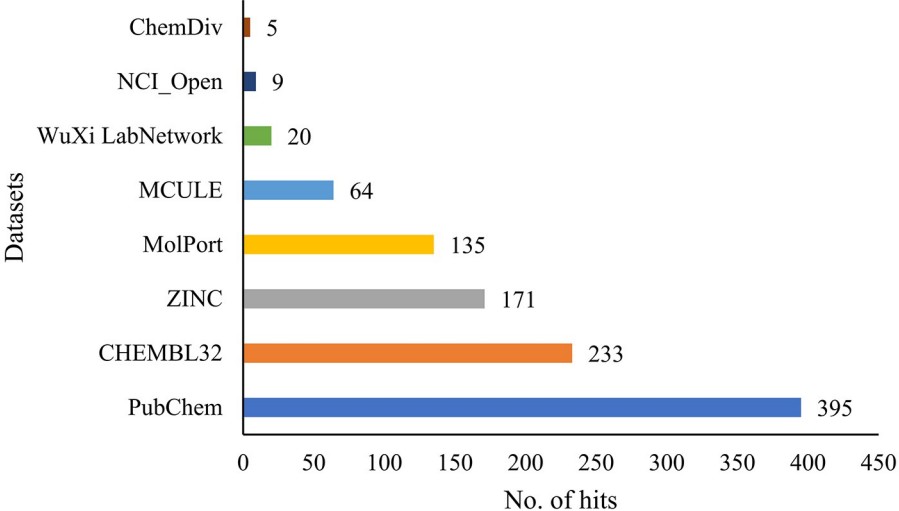

**Fig 2. The hit numbers are a result of PharmIt database screening.** It is worth noting that no hits were retrieved from ChemSpace, Enamin, and MCULE_ULTIMATE.

respectively. Furthermore, the Enrichment Factor (EF) measurements highlighted the model's effectiveness in detecting true positives, showing EF values 4.4 at the tested thresholds. Most importantly, the model showed zero false positives (FP), validating its accuracy and reliability in identifying active compounds. These findings show the strength and possible usefulness of the flavonol-based model in identifying compounds that match pharmacophores, providing essential information for future drug discovery efforts.

## The QSAR model

The QSAR model (see S1 File) was established to reliably predict the biological properties of the flavonols used in constructing the pharmacophore model. However, two observations were randomly separated to be used as the external validation set. Different models were created using varying numbers of random internal validation sets of the non-flavonol flavonoids or non-flavonoids. The best model with the highest adjusted $R^2$ was selected for the report. The chosen model was trained using Kaempferol, Isorhamnetin, Icaritin, Hexamethoxyflavone, Chrysoeriol-6-C-b-D-boivinopyranosyl-41-O-b-D-glucopyranoside, Hyperoside, and Q3G. Also, the random validation test sets consisted of matched numbers of flavonoids (Proanthocyanidin, Amentoflavone, Rosmarinic acid, Sakuranetin, OHBF-C, Luteolin 7-O-glucuronide, and Isovitexin) or non-flavonoids (PHAP, Nobiletin, Linalool, Betulinic acid, Ursolic acid, Taraxasterol, and Solamargine). This was due to the limited number of training sets and the fact that bias was prevented in the model.

Three sets of 2D descriptors were generated using RDKit, Dragon, and Mordred. After removing duplications and constant or near-constant variables, the final number of predictor variables was 759. The results of the variable selection of the modeling process are summarized in Table 1. Based on the Type III sum of squares, only two variables, the average connectivity index of order 4 (*X4A*) and the quantitative estimate of drug-likeness (*qed*), had significant information to explain the variability of the biological activities. Among the explanatory variables, based on the Type III sum of squares (Table 2), *X4A* was more influential. No multicollinearity was observed for these predictors (VIFs < 10). It was also observed that the same

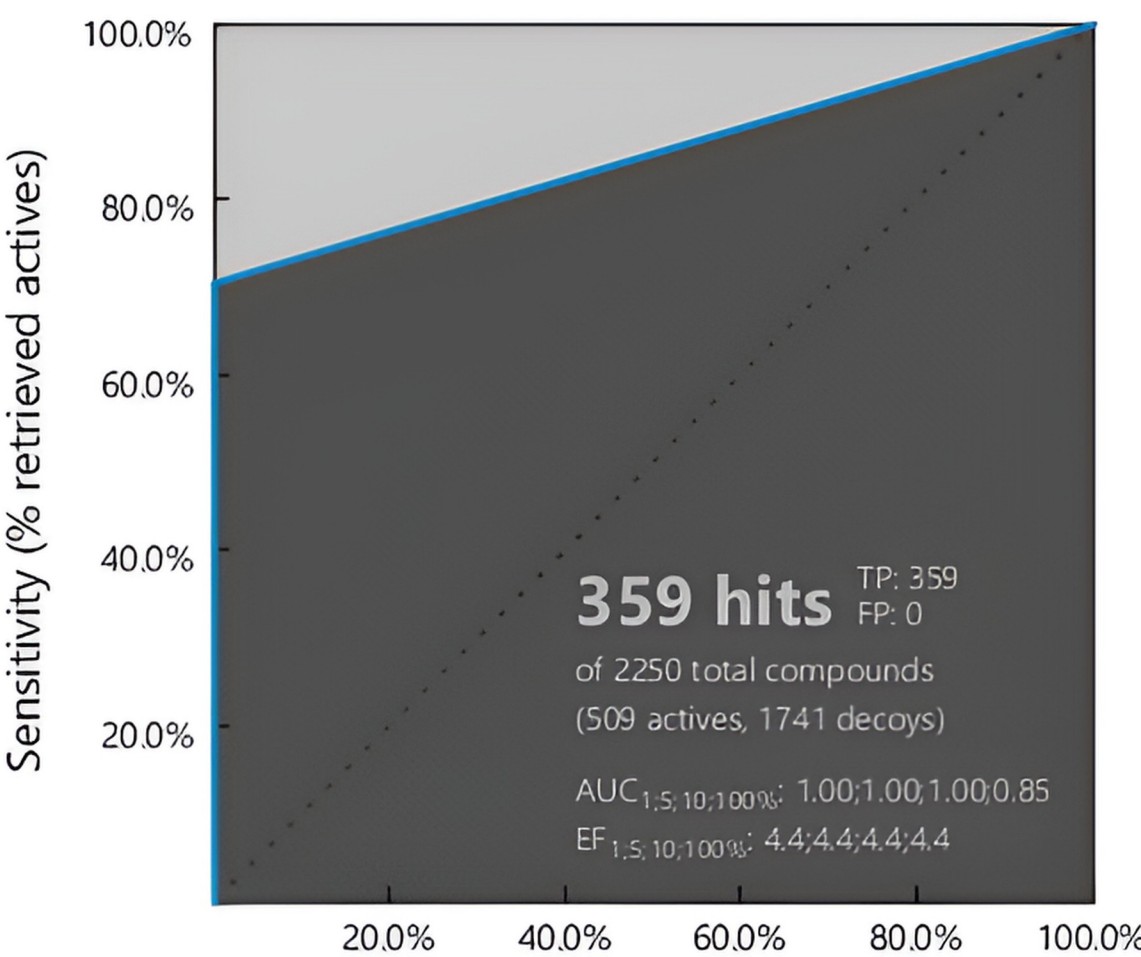

**Fig 3. ROC curve analysis of the flavonol-derived pharmacophore model.** 359/509 hits (70.53%) of hits were retrieved, suggesting the strength of the model to specify the potentially active compounds.

**Table 1. Summary of the variables selection to predict biological activities of the flavonoids with anti-HBV activities.**

| Nbr. of variables | Variables | MSE | $R^2$ | Adjusted $R^2$ | Akaike's AIC | Schwarz's SBC | Amemiya's PC |
|---|---|---|---|---|---|---|---|
| 2 | X4A / qed | 102.80 | 0.87 | **0.85** | 67.48 | 69.40 | 0.17 |

The best model for the selected selection criterion is displayed in italic

**Table 2. Type III sum of squares analysis (biological activities).**

| Source | DF | Sum of squares | Mean squares | F | Pr > F |
|---|---|---|---|---|---|
| X4A | 1 | 5610.79 | 5610.79 | 54.58 | **< 0.0001** |
| qed | 1 | 3280.34 | 3280.34 | 31.91 | **0.00** |

**Table 3. Goodness of fit statistics (biological activities).**

| Non-flavonoid test set | | | Flavonoid validation test set | | |
|---|---|---|---|---|---|
| Statistic | Training set | Validation set | Statistic | Training set | Validation set |
| Observations | 14.00 | 7.00 | Observations | 14.00 | 7.00 |
| Sum of weights | 14.00 | 7.00 | Sum of weights | 14.00 | 7.00 |
| DF | 11.00 | 4.00 | DF | 11.00 | 4.00 |
| $R^2$ | 0.87 | 0.00 | $R^2$ | 0.87 | 0.03 |
| Adjusted $R^2$ | 0.85 | | Adjusted $R^2$ | 0.85 | |
| MSE | 102.80 | 217498.45 | MSE | 102.80 | 13862.97 |
| RMSE | 10.14 | 466.37 | RMSE | 10.14 | 117.74 |
| MAPE | 28.44 | 2971.25 | MAPE | 28.44 | 4965.21 |
| DW | 1.19 | | DW | 1.19 | |
| AIC | 67.48 | | AIC | 67.48 | |
| SBC | 69.40 | | SBC | 69.40 | |
| PC | 0.20 | | PC | 0.20 | |
| Press | 838.87 | | Press | 838.87 | |
| $Q^2$ | 0.90 | 0.00 | $Q^2$ | 0.90 | 0.00 |

predictors are adequate for establishing the model with non-flavonol flavonoids as the internal validation test set or non-flavonoid compounds. This suggests that the same variables distinguish flavonols from other compounds and determine the model's specificity.

The fitness and performance of the selected model were also further measured in training- and test sets by Goodness-of-fit statistics (see Table 3). Accordingly, the model achieved an $R^2$ value of 0.87 for the training set. The Adjusted $R^2$ value of 0.85 further confirmed the model's robustness, indicating that including descriptors appropriately balances model complexity and goodness of fit. As a finding, the model's error metrics were small, suggesting a relatively good accuracy of the trained model in predicting the observations. Also, $Q^2$ (0.9) was significantly high, indicating a good prediction power of the model on the external prediction set (see AD analysis results). Furthermore, the model error parameters in the validation sets differed interestingly. MSE, RMSE, and MAPE were substantially lower in the model validated by the non-flavonol flavonoids than that validated by non-flavonoids. This suggests the association between all flavonoids and the discriminative power of the model on the flavonol compounds.

Further analysis of variance (ANOVA) results showed that the model is statistically significant (95%CI F = 36.91 df(2), p < 0.0001), indicating that the selected descriptors collectively have a substantial impact on the biological activities of the compounds. The model explains a significant amount of the variability in the data, as indicated by the high sum of squares associated with the model (7589.40). The model over fitness was checked by the standardized coefficients measured for the variables in the mode (Table 4). Furthermore, the robustness of the predictive strength of the model was evaluated by the analysis of residuals (Tables 5 and 6).

**Table 4. Standardized coefficients of predictive variables in the QSAR model.**

| Validation set | Descriptor | Coefficient | Std. Error | t-value | P-value | Lower Bound | Upper Bound |
|---|---|---|---|---|---|---|---|
| Flavonoid | X4A | 0.82 | 0.03 | 25.93 | < 0.0001 | 0.75 | 0.88 |
| | qed | -0.62 | 0.03 | -19.17 | < 0.0001 | -0.70 | -0.55 |
| Non-flavonoid | X4A | 0.82 | 0.03 | 25.93 | < 0.0001 | 0.75 | 0.88 |
| | qed | -0.62 | 0.03 | -19.17 | < 0.0001 | -0.70 | -0.55 |

**Table 5. Predictions and residuals (biological activities) from the model validated by non-flavonol flavonoids.**

| Observation | Biological activities | Pred(Biological activities) | Residual | Std. residual | Std. dev. on pred. (Mean) | Adjusted Pred. |
|---|---|---|---|---|---|---|
| Kaempferol | 34.96 | 48.75 | -13.79 | -1.36 | 4.22 | 51.64 |
| Isorhamnetin | 63.28 | 46.64 | 16.64 | 1.64 | 4.36 | 42.87 |
| Icaritin | 10.00 | 5.73 | 4.27 | 0.42 | 4.65 | 4.59 |
| Hexamethoxyflavone | 11.37 | 19.13 | -7.76 | -0.77 | 3.83 | 20.42 |
| Chrysoeriol-6-C-b-D-boivinopyranosyl-41-O-b-D-glucopyranoside | 83.26 | 82.51 | 0.75 | 0.07 | 6.11 | 82.08 |
| Hyperoside (quercetin-3-O-galactoside) | 32.30 | 35.47 | -3.17 | -0.31 | 4.13 | 36.10 |
| Quercetin-3-O-glucuronide_(Q3G) | 26.13 | 23.08 | 3.05 | 0.30 | 5.16 | 22.01 |
| Proanthocyanidin | 20.00 | -11.17 | 31.17 | 3.07 | 9.65 | -11.17 |
| Amentoflavone | 90.20 | 26.77 | 63.43 | 6.26 | 5.45 | 26.77 |
| Rosmarinic acid | 30.00 | 249.24 | -219.24 | -21.62 | 27.91 | 249.24 |
| Sakuranetin | 43.69 | 60.05 | -16.36 | -1.61 | 10.03 | 60.05 |
| Oolonghomobisflavan C (OHBF-C) | 4.30 | 35.25 | -30.95 | -3.05 | 6.31 | 35.25 |
| Luteolin 7-O-glucuronide | 55.51 | 72.27 | -16.76 | -1.65 | 4.89 | 72.27 |
| Isovitexin | 0.09 | 29.80 | -29.71 | -2.93 | 3.69 | 29.80 |

**Table 6. Predictions and residuals (biological activities) from the model validated by non-flavonoids.**

| Observation | Biological activities | Pred(Biological activities) | Residual | Std. residual | Std. dev. on pred. (Mean) | Adjusted Pred. |
|---|---|---|---|---|---|---|
| Kaempferol | 34.96 | 48.75 | -13.79 | -1.36 | 4.22 | 51.64 |
| Isorhamnetin | 63.28 | 46.64 | 16.64 | 1.64 | 4.36 | 42.87 |
| Icaritin | 10.00 | 5.73 | 4.27 | 0.42 | 4.65 | 4.59 |
| Hexamethoxyflavone | 11.37 | 19.13 | -7.76 | -0.77 | 3.83 | 20.42 |
| Chrysoeriol-6-C-b-D-boivinopyranosyl-41-O-b-D-glucopyranoside | 83.26 | 82.51 | 0.75 | 0.07 | 6.11 | 82.08 |
| Hyperoside (quercetin-3-O-galactoside) | 32.30 | 35.47 | -3.17 | -0.31 | 4.13 | 36.10 |
| Quercetin-3-O-glucuronide_(Q3G) | 26.13 | 23.08 | 3.05 | 0.30 | 5.16 | 22.01 |
| PHAP | 73.50 | 405.98 | -332.48 | -32.79 | 51.55 | 405.98 |
| Nobiletin | 33.90 | 6.23 | 27.67 | 2.73 | 4.59 | 6.23 |
| Linalool | 7.10 | 765.39 | -758.29 | -74.79 | 100.32 | 765.39 |
| Betulinic acid | 50.00 | -213.33 | 263.33 | 25.97 | 33.84 | -213.33 |
| Ursolic acid | 7.10 | -198.67 | 205.77 | 20.29 | 32.05 | -198.67 |
| Taraxasterol | 56.29 | -196.63 | 252.92 | 24.95 | 31.98 | -196.63 |
| Solamargine | 1.57 | -88.09 | 89.66 | 8.84 | 19.57 | -88.09 |

The fitness plot (Fig 4a & 4c) also shows the best-fitted linear regression of predicted observations within the training set. Comparing models validated with non-flavonol flavonoids and non-flavonoids highlights unique features in the linearity and fitness of their training sets. The model tested with chemicals that are not flavonoids shows a more substantial level of linearity and fitness (Fig 4b) when compared to the training set. However, the model confirmed that using flavonoids also shows significant fitting ability. These results emphasize the importance of using distinct validation sets to investigate structural variations in flavonoids thoroughly. Furthermore, the Cook's distance metrics (Fig 4b) were checked, and no outliers were observed (Cook's distance < 1).

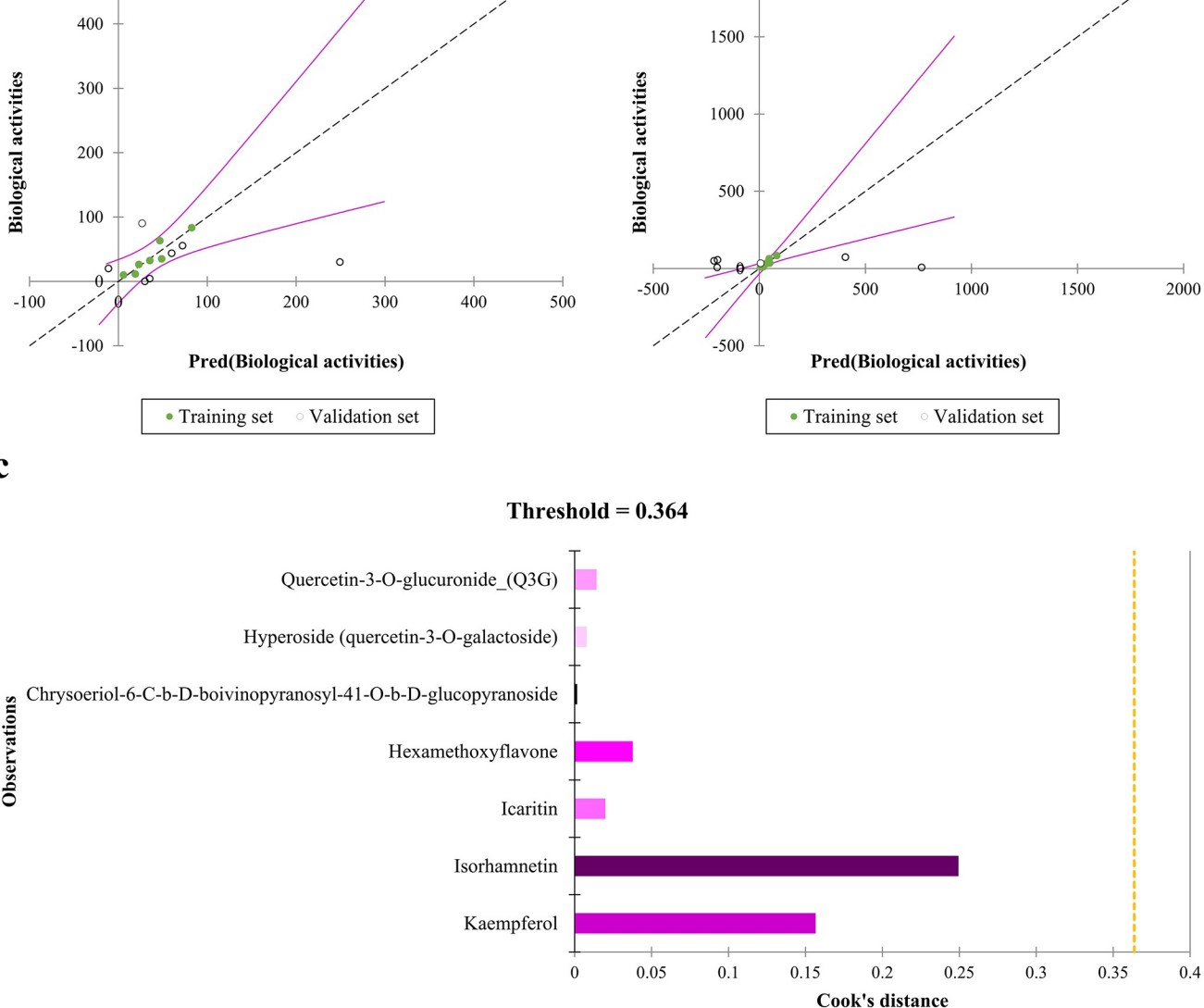

**Fig 4. Fitness and Cook's distance plots of actual biological activities vs predicted values.** Panels **a** & **b** show the fitness plots for QSAR models validated with non-flavonol flavonoids and non-flavonoids, respectively. The result demonstrated that more flavonols with anti-HBV activities will strengthen the model validated with flavonoids. Cook's distance metric is also depicted in panel **c**.

## Applicability Domain (AD) analysis

AD analysis was performed to determine the range of chemical space where the QSAR model can make reliable predictions and to identify the boundaries of the training data in which predicted biological activities of the external set fall within, ensuring that the model is applicable to predict the same compounds. Unfortunately, the number of flavonols was limited. Of nine compounds, seven were randomly used to train the QSAR model, and the rest were used to calculate Euclidian distance (d) and determine the model's AD. Two flavonols, Quercetin and Myricetin 3-rhamnoside, were used as an external validation set. The compounds' predicted

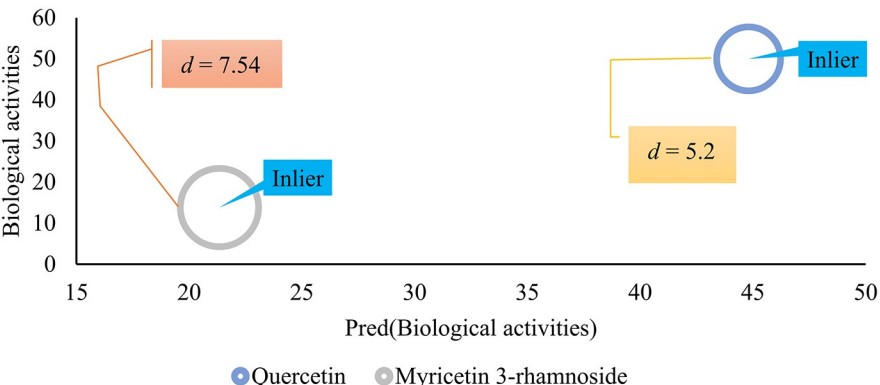

**Fig 5. AD of the predicted external validation set.** The size of the circle is based on the calculated d.

biological activities were measured in terms of distance, and if it fell within the specified threshold, it was considered an inlier, indicating the model's applicability (Fig 5).

The mean calculated *d* for the training set was 7.06 ± 5.56. As described above, the threshold was measured as 2x St.dev plus mean *d*. Accordingly, the threshold was set as 18.178. As shown in Fig 3, the calculated *d* both compounds were *inlier*, indicating the model's applicability for other flavonols with anti-HBV activities.

## PCA measure of the predictors

The principal component analysis (PCA) was conducted on the training set's variables, including *Biological activities*, *X4A*, *qed*, and *MW* as a supplementary dummy variable. According to the Pearson correlation matrix, *Biological activities* exhibited a strong positive correlation with X4A (r = 0.70) and a moderate negative correlation with qed (r = -0.48). Bartlett's sphericity test indicated that the variables were not independent (Chi-square = 8.65, p = 0.03), supporting the significance of the correlation between variables and the need for PCA analysis. The eigenvalues of the components revealed that the first three components (F1 to F3) explained 59.09%, 38.89%, and 2.02% of the total variability, respectively. The cumulative percentage of variance explained by the first two components was 97.98% (Fig 6), suggesting that these components capture most of the dataset's variability.

It was found that the first component (F1) had a significant loading for *Biological activities* (0.74), *X4A* (0.60), and *qed* (-0.32), indicating their strong influence on this component. The second component (F2) exhibited significant loadings for *X4A* (0.55), *qed* (0.83), and *MW* (-0.80), suggesting their association with this dimension (see Fig 7a). In contrast, the third component (F3) had a minor eigenvalue of 0.06, indicating its marginal role in explaining the dataset's variance. Considering the scree plot method and the Kaiser criterion, the analysis supports retaining the first two principal components. These components, F1 and F2, cumulatively explain nearly 98% of the total variance in the dataset, making them suitable for further analysis and interpretation. Therefore, based on the proportion of variance explained and eigenvalues, these two PCs were retained to capture the essential patterns and relationships within the data.

As shown in Fig 7b, PC1 primarily measures compounds' bioactivity. The second principal component (PC2) shows large positive associations with qed (0.90) and X4A (0.59) while negatively associated with MW (-0.80). PC2 primarily measures compounds' qed and structural complexity (X4A).

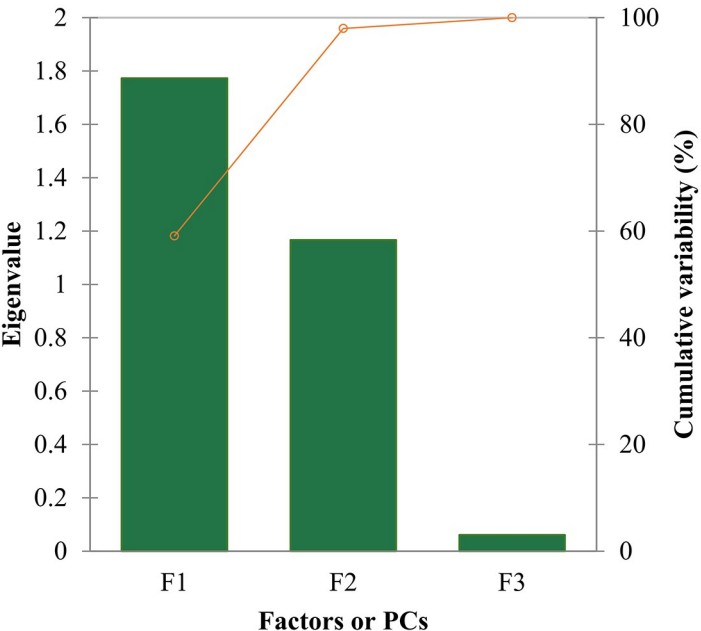

**Fig 6. Scree plot.** The whole variance of the model's data is explained in three factors. Accordingly, ~98% of data are displayed with two first F1 and F2 components that are used to explain the variables of the QSAR model.

As shown in Fig 7c, the factor scores from the PCA revealed distinct patterns among the flavonols. F1 showed a positive association with Kaempferol, Isorhamnetin, and Chrysoeriol-6-C-b-D-boivinopyranosyl-41-O-b-D-glucopyranoside, indicating these compounds have higher scores on this factor. In contrast, Factor 2 is negatively associated with Chrysoeriol-6-C-b-D-boivinopyranosyl-41-O-b-D-glucopyranoside, hyperoside, and Q3G, suggesting these compounds have lower scores on this factor. This pattern was further confirmed with Bootstrap sampling analysis with a minor variation (Fig 7d).

### Integration of the QSAR model on the screened flavonols

Similarly, *qed* and *X4A* were predicted for the screened compounds to predict their $IC_{50}$ ($pIC_{50}$). The integration of the QSAR model with the top 10 screened flavonol hits is shown in Fig 8A. The $pIC_{50}$ values for the hits ranged from 23.08 nM to 141.68 nM, comparable to the IC50 values of the compound used in model validation. The results indicated a non-normal distribution of calculated rmsd and the number of conformers for the selected hits (Fig 8B). However, there was a positive linear correlation between $pIC_{50}$ and *X4A* and a negative linear correlation between $pIC_{50}$ and *qed* (Fig 8D). Also, similar correlation findings were observed between *X4A* (Pearson's r = 0.620, *p*-value $<$ 0.05) and *qed* (Pearson's r = -0.627, *p*-value $<$ 0.05) with the number of conformers. The STRINGS of the selected compounds are also demonstrated in Fig 8C. Furthermore, the calculated $pIC_{50}$ is also provided in the S2 File.

### Discussion

The development of effective therapeutic interventions against HBV infection is important, given its significant global health burden and potential for severe complications. Flavonoids, a broad range of natural substances present in many plants, are being studied for their possible antiviral effects, such as their capacity to disrupt different phases of the HBV lifecycle [27]. The

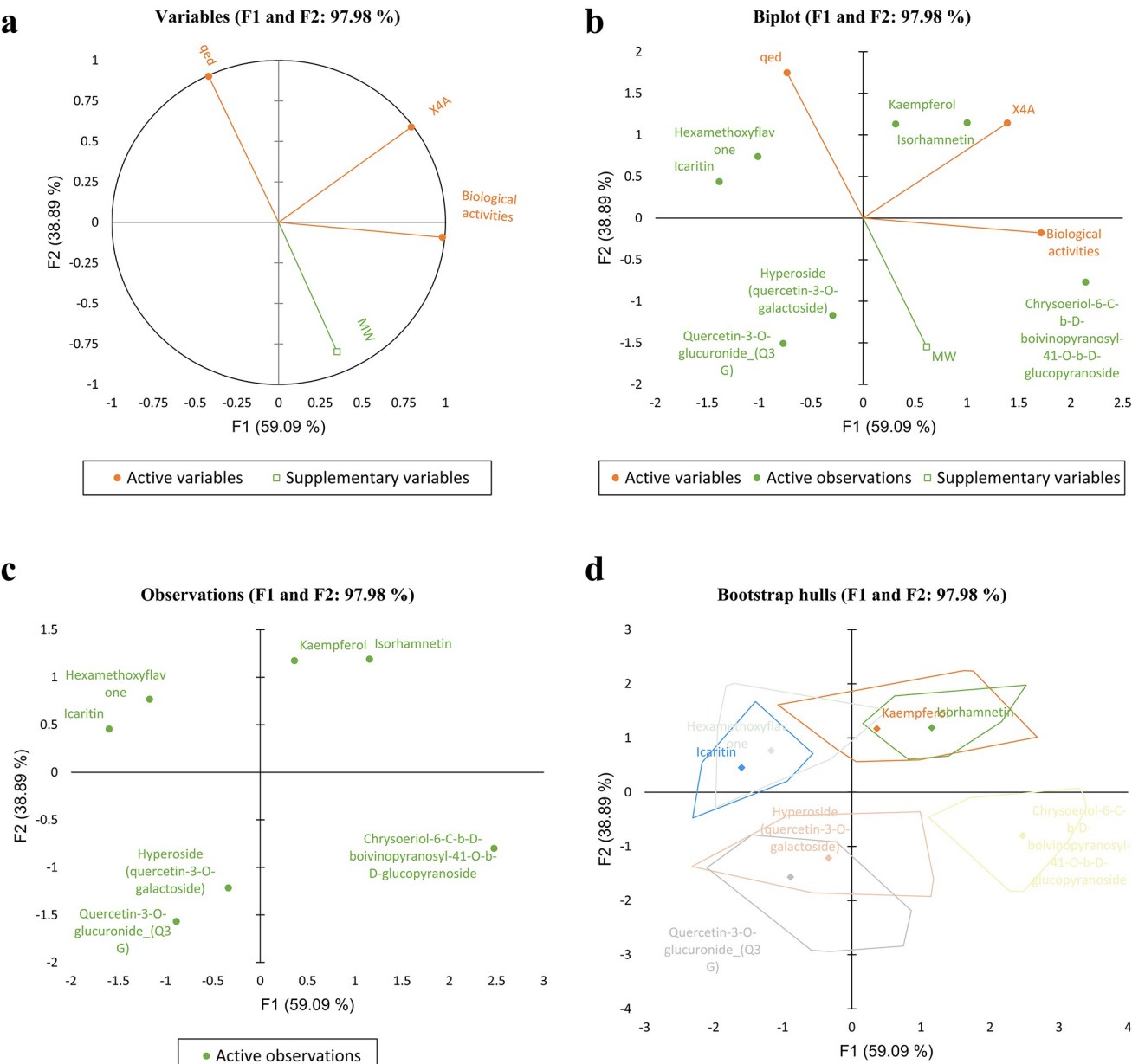

**Fig 7. The PCA plots representing the variables relationships (a & b), and the score plots of distribution patterns (c) and variation of compounds (d) in two PCs.**

variety of flavonoids offers an interesting chance to find new antiviral drugs, leading to the creation of models based on pharmacophores and QSAR to help identify and predict compounds with strong anti-HBV effects through virtual HTS. Previous studies have also highlighted the potential of utilizing computational modeling like QSAR to offer potential compounds with anti-HBV activities [75–78]. The present study adds to our knowledge of flavonoid pharmacology and has implications for creating new antiviral treatments and finding plants with high levels of bioactive flavonoids.

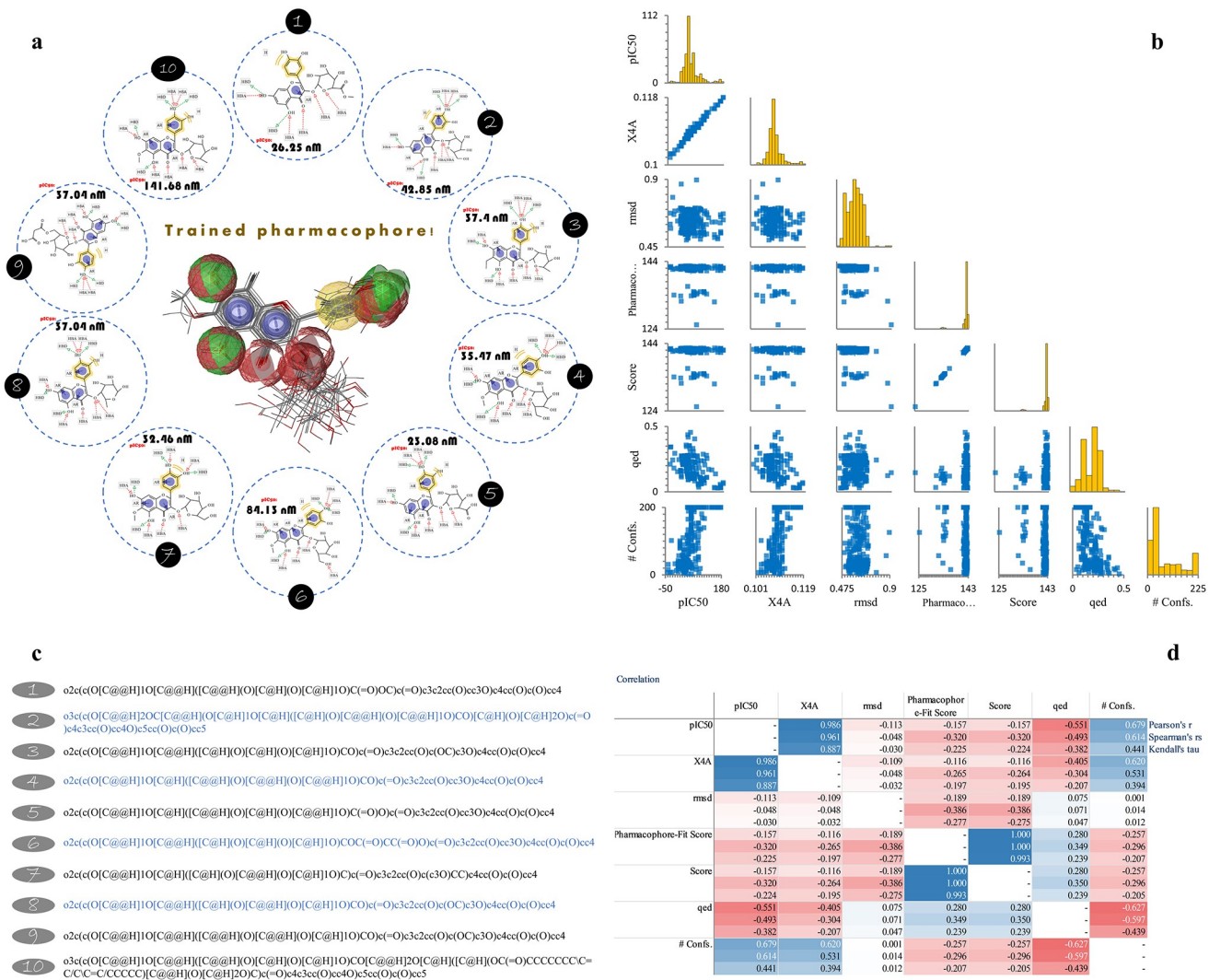

**Fig 8. Alignment of top 10 identified flavonol hits with the trained pharmacophore model.** This figure shows the chemical structures of the top 10 identified flavonol hits (A), represented by their SMILES strings (C), aligned with the trained pharmacophore model. The pair or histogram plot (B) demonstrates a non-normal distribution of rmsd and the number of conformers of the hits. The score and pharmacophore-hit score are correlated, which was not surprising (D).

The development and validation of pharmacophore models are essential in contemporary drug discovery endeavors [35, 69, 79–81]. This study carefully developed a validated pharmacophore model based on flavonols, consisting of 57 features such as hydrophobic centers, aromatic rings, HBA and HBD regions, and exclusive volumes. The accuracy of the top model showed its potential to precisely screen compounds with pharmacophore characteristics related to flavonols. In this regard, The ROC curve analysis showed encouraging performance metrics, including a sensitivity of 71% and a specificity of 100%. The AUC values also confirmed the discriminative ability of the model. The model is possibly highly effective in narrowing down candidate compounds because of its high AUC values, which suggest minimal false positives. This accuracy in screening aids in directing experimental validation towards top-notch hits, thereby speeding up the drug discovery process. With the sensitivity and specificity demonstrated, the model is a strong tool in searching for more effective HBV treatments

by uncovering new flavonol-based compounds. Accordingly, a curated collection of 359 distinct compounds was highlighted as active true positive chemicals with potential anti-HBV activities. These findings provide an excellent opportunity for vast research on these compounds *in vitro*. Also, since the training set of flavonols showed different anti-HBV mechanisms of action, the hits resulting from the screening potentially can display different anti-HBV activities. Also, future efforts to discover such compounds will provide further chances for developing a more sensitive model. The screening results were not the focus of this study, yet the required information can be found in the (S2 File).

The QSAR model was constructed to predict the biological properties of flavonols utilized in the pharmacophore model. Noteworthy is the model's level of predictive power with no sign of collinearity or overfitting penalties, achieving an $R^2$ value of 0.87 in the training dataset. The model's explanatory capability was significantly affected by the predictors *X4A* and *qed*. The model's robust performance was further confirmed by its ability to generalize well to the external validation set, demonstrated by the 90% $Q^2$ value and *inlier* AD calculated for the external validation set. The developed model benefits from the biological activity of compounds in inhibiting the secretion of HBsAg. Therefore, the prediction applies to flavonoids with anti-HBsAg activities. This provides a further opportunity to investigate and reconstruct another model to search for variables predictive of the biological activities of flavonols on another aspect of viral infection, like HBeAg. In the present study, modeling different QSARs was impossible due to the limited number of flavonols affecting another part of viral components. The presented model was reevaluated with different input variables and other affirmatory statistics and revalidated with a different number of validation sets, yet the results were the same. In this regard, the validated models based on non-flavonol flavonoids and non-flavonoids are reported for comparison. Accordingly, one intriguing finding was the varying success of the model when tested with non-flavonol flavonoids compared to non-flavonoids. Although both models demonstrated good fit, the non-flavonoid model displayed more robust linearity and fitness with the training data. This implies that the model can accurately differentiate between flavonoids and other compounds by analyzing their structural characteristics. Nevertheless, the model verified with flavonoids also showed significant predictive ability, emphasizing the distinct structural differences in flavonol substances. This might be attributed to the predictors. The average connectivity index of order 4 (*X4A*) is a topological index. It has been shown to have an influence on the binding affinities of Per- and poly-fluoroalkyl substances (PFAS) ligands to Human serum albumin (HSA) receptor and is a significantly influential score in establishing QSAR model [77]. In a QSAR study by Hădărugă [82], *X4A* was identified as a significant structural parameter related to the inhibitory action of flavonoids against cytochrome P450. The study found that the biological activity increased with the decrease in the X4A parameter value. Specifically, higher *X4A* values were associated with lower inhibitory activity against cytochrome P450. This indicates that X4A plays a role in describing the hydrophilic-hydrophobic properties or polarizabilities of the flavonoid molecules in relation to their inhibitory activity against cytochrome P450. These suggest that X4A is a major contributor to the topological structure of flavonoids that facilitate the binding of the chemical to the target. Here, the PCA results showed a significant positive correlation between the biological activity of compounds with X4A, suggesting the increase in X4A will enhance flavonols' $IC_{50}$. Another affected predictor was a quantitative estimate of drug-likeness (*qed*). It was found that *qed* has an orthogonal association with biological activity, the variables are not redundant, and each contributes unique information. *qed* is a measure used to assess the drug-likeness of chemical compounds and combines several molecular properties associated with drug-like behavior into a single score. The mean *qed* calculated for flavonols was 0.41 ± 0.19, highlighting a minor trend toward druglike compounds. Accordingly, Lee

et al. showed that flavonoids with $qed \geq 0.35$ are more likely to exhibit drug-like behavior, making them promising candidates for studies related to drug development or therapeutic effects [83]. Additionally, it was observed that establishing a QSAR model with the $qed$ score provides an efficient approach to discovering flavonoid derivatives with potent anti-dipeptidyl peptidase-IV (DPP-IV) activity [84]. The features selected for developing a QSAR model in the present study are confirmed by the previous studies, suggesting that both structural and drug-likeness properties of flavonoids strengthen the predictive power of the models.

While this work sheds light on the potential of flavonol-based pharmacophore models and QSAR analysis in anti-HBV drug development, several significant drawbacks exist. One such constraint is the possibility of bias in the screening method, as the HTS library was predominantly focused on flavonoids. This method may not capture the entire chemical variety of flavonoids or other possible anti-HBV chemicals. Furthermore, focusing on a single subclass limits the model's applicability to other subclasses with distinct structural properties and action methods. Also, only flavonols were used to train both pharmacophore and QSAR models. This contributed to advancing the existing understanding of flavonols' effectiveness against HBV; however, utilizing models trained with different subclasses of flavonoids could be essential. Future research should include increasing the training set to include a larger variety of flavonoid subclasses and investigating other varied chemical libraries to improve the model's resilience and predictive potential. We are currently studying models trained with flavonoids for uncovering anti-HBV drugs, mainly using machine learning algorithms other than MLR to assess the accuracy of the models. The current research results also offer a way to identify plants that contain flavonoids with anti-HBV properties by tracking the screened hits.

## Conclusion

Druglike flavonoids with anti-HBV activities are possessed a yet-to-discovery potential. The present study validated an accurate flavonol-based pharmacophore model for screening large chemical libraries, highlighting novel hits for further experimental studies. Also, a QSAR model was validated by two independent sets of chemicals for reproducibility of the model. In addition, the model applied to other flavonols due to the predictive selected features, $x4a$ and $qed$. Findings pave the way for future anti-HBV drug discovery modeling and experimental studies. Discoveries provide a promising path for future modeling and experimental research on anti-HBV drug development. This study built upon existing research on flavonoids as potential treatments for HBV, showing promising prospects for future HBV therapeutics.

## Supporting information

**S1 File. Raw data and results of the QSAR equation.** Data includes training sets with their respective predicted affinities (dependent variables), 2D descriptors (independent variables), and applicability domain calculation.
(XLSX)

**S2 File. The screening results.** The file contains names of the hits, smiles, Mol. Index, Active/Decoy, Source Database, #Confs, Pharmacophore Match, Score, Pharmacophore-Fit Score, and rmsd.
(XLSX)

## Author Contributions

**Data curation:** Basireh Baei, Parnia Askari, Fatemeh Sana Askari.

**Formal analysis:** Basireh Baei, Fatemeh Sana Askari.

**Investigation:** Parnia Askari, Seyed Jalal Kiani.

**Methodology:** Alireza Mohebbi.

**Project administration:** Alireza Mohebbi.

**Resources:** Seyed Jalal Kiani, Alireza Mohebbi.

**Software:** Basireh Baei, Parnia Askari, Alireza Mohebbi.

**Supervision:** Alireza Mohebbi.

**Validation:** Seyed Jalal Kiani, Alireza Mohebbi.

**Visualization:** Alireza Mohebbi.

**Writing – original draft:** Basireh Baei, Parnia Askari, Fatemeh Sana Askari, Seyed Jalal Kiani, Alireza Mohebbi.

**Writing – review & editing:** Alireza Mohebbi.

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
