## [Decision Letter · Decision Letter 0]

9 Sep 2024

PONE-D-24-19041Ligand-based pharmacophore modeling, virtual screening, and 2D quantitative structure-activity relationship performance on anti-Hepatitis B virus flavonolsPLOS ONE

Dear Dr. Mohebbi,

Thank you for submitting your manuscript to PLOS ONE. After careful consideration, we feel that it has merit but does not fully meet PLOS ONE’s publication criteria as it currently stands. Therefore, we invite you to submit a revised version of the manuscript that addresses the points raised during the review process.

We look forward to receiving your revised manuscript.

Kind regards,

Nafees Ahemad

Academic Editor

PLOS ONE

Journal Requirements:

2. In the online submission form, you indicated that [The data produced and analyzed in the present study are included in the paper and are also available in the supporting Information. Moreover, the corresponding author can supply additional details upon request.]. 

Reviewers' comments:

Reviewer's Responses to Questions

**Comments to the Author**

1. Is the manuscript technically sound, and do the data support the conclusions?

Reviewer #1: Yes

Reviewer #2: Yes

2. Has the statistical analysis been performed appropriately and rigorously? 

Reviewer #1: Yes

Reviewer #2: Yes

3. Have the authors made all data underlying the findings in their manuscript fully available?

Reviewer #1: Yes

Reviewer #2: Yes

4. Is the manuscript presented in an intelligible fashion and written in standard English?

Reviewer #1: Yes

Reviewer #2: No

5. Review Comments to the Author

Reviewer #1: Authors of the presented manuscript explored the anti-Hepatitis B potentiality of flavonol-based metabolites through combined ligand-based approaches. The study is considered relevant within the field of drug discovery. Few comments and suggestions are to be highlighted.

1. The abstract is better presented as single paragraph.

2. Authors should elaborate more on validating the pharmacophoric model through using more than one validation approach; cost analysis, decoy set and/or Fischer's method.

3. Chemical structures or even the SMILES strings of the identified hits should be presented. Additionally, aligning the identified hits with the deduced pharmacophoric model should be presented.

4. Finally, concerning the discussion/conclusion, authors are advised to elaborate more on the future of this work? What are the study limitations and what approaches could be conducted to further address them?

Reviewer #2: Title : It could be more concise e.g."Pharmacophore Modeling and QSAR Analysis of Anti-HBV Flavonols."

Abstract: Include a brief statement about the study's significance in relation to current HBV treatment options.

Methods: Certain technical terms may require clarification for readers who are not well-versed in the field. For example, a brief explanation of "Euclidean distance calculations" would be helpful.

Results: Some results lack adequate context or explanation. For example, when discussing the ROC curve, it would be beneficial to clarify what the specific AUC values indicate about the model's effectiveness. It is also suggested that the authors should provide a more thorough interpretation of the results, particularly regarding their implications for drug discovery.

Discussion: While the strengths of the models are highlighted, it is also important to address potential limitations and areas for improvement. I suggest that a paragraph discussing the study's limitations, such as possible biases in the screening process or the limited diversity of the flavonoids examined should be added.

General Comments

Strengths:

The manuscript conducts a comprehensive investigation into the pharmacological potential of flavonols against HBV, offering valuable insights to the field. However, there are grammatical errors and awkward phrasing throughout the text and a thorough proofreading is therefore highly recommended to identify any remaining errors and improve overall clarity.

Conclusion

The study presents significant findings that could enhance the understanding of anti-HBV compounds and contribute to the development of new therapeutic strategies. But focusing on clarity, critical analysis of results, and comprehensive proofreading will improve the manuscript's overall quality.

Recommendation

I consider it appropriate for publication in PLoS ONE, provided the authors address the comments and suggestions mentioned above.

6. PLOS authors have the option to publish the peer review history of their article (what does this mean?). If published, this will include your full peer review and any attached files.

Reviewer #1: **Yes: **Khaled M Darwish

Reviewer #2: No

---

## [Author Response · Author response to Decision Letter 0]

21 Sep 2024

September 19, 2024

PONE-D-24-19041

Title: Ligand-based pharmacophore modeling, virtual screening, and 2D quantitative structure-activity relationship performance on anti-Hepatitis B virus flavonols

Journal: PLOS ONE

Response to Reviewers

Dear Dr. Nafees Ahemad,

We thank you for the opportunity to revise our manuscript and the valuable feedback during the review process. We have carefully considered all the comments and have made the necessary revisions to improve the manuscript. Below, we provide a detailed response to each reviewer's comment.

Journal Requirements:

Action taken: We have revised our manuscript to comply with PLOS ONE's style requirements, including appropriate file naming conventions.

2. In the online submission form, you indicated that [The data produced and analyzed in the present study are included in the paper and are also available in the supporting Information. Moreover, the corresponding author can supply additional details upon request.]. 

Action taken: We have ensured that all data underlying our study's findings are freely available. The data are included in the manuscript and supporting information files.

Action taken: We have reviewed our reference list to ensure it is complete and accurate

Reviewer #1 Comments and Responses:

1. Comment: The abstract is better presented as a single paragraph.

o Response: We have revised the abstract to present it as a cohesive paragraph. Additionally, we have included a statement about the study's significance in relation to current HBV treatment options to provide a more comprehensive overview.

2. Comment: Authors should elaborate more on validating the pharmacophoric model through using more than one validation approach; cost analysis, decoy set, and/or Fischer's method.

o Response: We have revised the method section in the revised manuscript. Accordingly, the Drugbank was used as a decoy to validate the PharmIt screening and ensure that the model was validated to yield flavonols potentially. The wording has been revised.

3. Comment: Chemical structures or even the SMILES strings of the identified hits should be presented. Additionally, aligning the identified hits with the deduced pharmacophoric model should be presented.

o Response: We have incorporated the identified hits' chemical structures and SMILES strings into the results section in a new Result section and a new Figure. Furthermore, we have updated the S2 Table by including the STRINGS and predicted biological activity based on the QSAR model for the screened flavonols.

4. Comment: Concerning the discussion/conclusion, authors are advised to elaborate more on the future of this work. What are the study limitations and what approaches could be conducted to further address them?

o Response: The discussion and conclusion sections have been expanded to address the future implications of this work, study limitations, and potential future approaches. We discuss the limitations, such as the limited diversity of flavonoids examined and possible biases in the screening process, and propose strategies for further research.

Reviewer #2 Comments and Responses:

1. Comment: Title: It could be more concise e.g., "Pharmacophore Modeling and QSAR Analysis of Anti-HBV Flavonols."

o Response: We have revised the title to "Pharmacophore Modeling and QSAR Analysis of Anti-HBV Flavonols" to make it more concise and reflective of the study's content.

2. Comment: Abstract: Include a brief statement about the study's significance in relation to current HBV treatment options.

o Response: A statement about the study's significance in the context of current HBV treatments has been added to the abstract to highlight the relevance and potential impact of the research findings.

3. Comment: Methods: Certain technical terms may require clarification for readers who are not well-versed in the field. For example, a brief explanation of "Euclidean distance calculations" would be helpful.

o Response: We have revised the methods section to briefly explain technical terms such as "Euclidean distance calculations" to ensure clarity for readers who may not be familiar with these concepts.

4. Comment: Results: Some results lack adequate context or explanation. For example, when discussing the ROC curve, it would be beneficial to clarify what the specific AUC values indicate about the model's effectiveness.

o Response: Additional context and explanation have been added to the results section, particularly regarding the ROC curve and AUC values, to clarify the model's effectiveness in identifying potential anti-HBV flavonols.

5. Comment: Discussion: While the strengths of the models are highlighted, it is also important to address potential limitations and areas for improvement. I suggest that a paragraph discussing the study's limitations, such as possible biases in the screening process or the limited diversity of the flavonoids examined, should be added.

o Response: We have added a paragraph in the discussion section addressing the study's limitations, including potential biases in the screening process and the limited chemical diversity of flavonoids examined. We also discuss areas for improvement and future research directions.

6. Comment: General Comments: There are grammatical errors and awkward phrasing throughout the text, and a thorough proofreading is therefore highly recommended to identify any remaining errors and improve overall clarity.

o Response: We have thoroughly proofread the manuscript, correcting grammatical errors and awkward phrasing to improve overall clarity and readability.

We believe these revisions have addressed the reviewers' comments and significantly improved the manuscript. We hope that the revised version meets the standards of PLOS ONE and look forward to your favorable consideration.

Thank you for your time and the opportunity to revise our manuscript.

Sincerely,

Alireza Mohebbi

Department of Virology, School of Medicine, Iran University of Medical Sciences, Tehran, Iran

Tel: +98 935 467 4593; Email: Alirezaa2s@gmail.com

---

## [Decision Letter · Decision Letter 1]

17 Dec 2024

Pharmacophore Modeling and QSAR Analysis of Anti-HBV Flavonols

PONE-D-24-19041R1

Dear Dr. Mohebbi,

We’re pleased to inform you that your manuscript has been judged scientifically suitable for publication and will be formally accepted for publication once it meets all outstanding technical requirements.

Kind regards,

Nafees Ahemad

Academic Editor

PLOS ONE

Additional Editor Comments (optional):

Reviewers' comments:

Reviewer's Responses to Questions

**Comments to the Author**

1. If the authors have adequately addressed your comments raised in a previous round of review and you feel that this manuscript is now acceptable for publication, you may indicate that here to bypass the “Comments to the Author” section, enter your conflict of interest statement in the “Confidential to Editor” section, and submit your "Accept" recommendation.

Reviewer #1: (No Response)

Reviewer #2: All comments have been addressed

2. Is the manuscript technically sound, and do the data support the conclusions?

Reviewer #1: (No Response)

Reviewer #2: Yes

3. Has the statistical analysis been performed appropriately and rigorously? 

Reviewer #1: (No Response)

Reviewer #2: Yes

4. Have the authors made all data underlying the findings in their manuscript fully available?

Reviewer #1: (No Response)

Reviewer #2: Yes

5. Is the manuscript presented in an intelligible fashion and written in standard English?

Reviewer #1: (No Response)

Reviewer #2: Yes

6. Review Comments to the Author

Reviewer #1: (No Response)

Reviewer #2: (No Response)

7. PLOS authors have the option to publish the peer review history of their article (what does this mean?). If published, this will include your full peer review and any attached files.

Reviewer #1: **Yes: **Khaled M. Darwish

Reviewer #2: **Yes: **Professor Olorunfemi A. Eseyin

---

## [Editor Report · Acceptance letter]

22 Dec 2024

PONE-D-24-19041R1 

PLOS ONE

Dear Dr. Mohebbi, 

I'm pleased to inform you that your manuscript has been deemed suitable for publication in PLOS ONE. Congratulations! Your manuscript is now being handed over to our production team.

Kind regards, 

on behalf of

Dr. Nafees Ahemad 

Academic Editor

PLOS ONE